# Pacing Dynamics Determines the Arrhythmogenic Mechanism of the CPVT2-Causing CASQ2^G112+5X^ Mutation in a Guinea Pig Ventricular Myocyte Computational Model

**DOI:** 10.3390/genes14010023

**Published:** 2022-12-22

**Authors:** Roshan Paudel, Mohsin Saleet Jafri, Aman Ullah

**Affiliations:** 1School of Systems Biology, George Mason University, Fairfax, VA 22030, USA; 2School of Computer, Mathematical, and Natural Sciences, Morgan State University, Baltimore, MD 21251, USA; 3Center for Biomedical Engineering and Technology, University of Maryland School of Medicine, Baltimore, MD 20201, USA

**Keywords:** heart, CPVT2, arrhythmia, CASQ2, computational model, alternans, early afterdepolarization, beat skipping, β-adrenergic

## Abstract

Calsequestrin Type 2 (CASQ2) is a high-capacity, low-affinity, Ca^2+^-binding protein expressed in the sarcoplasmic reticulum (SR) of the cardiac myocyte. Mutations in CASQ2 have been linked to the arrhythmia catecholaminergic polymorphic ventricular tachycardia (CPVT2) that occurs with acute emotional stress or exercise can result in sudden cardiac death (SCD). CASQ2^G112+5X^ is a 16 bp (339–354) deletion CASQ2 mutation that prevents the protein expression due to premature stop codon. Understanding the subcellular mechanisms of CPVT2 is experimentally challenging because the occurrence of arrhythmia is rare. To obtain an insight into the characteristics of this rare disease, simulation studies using a local control stochastic computational model of the Guinea pig ventricular myocyte investigated how the mutant CASQ2s may be responsible for the development of an arrhythmogenic episode under the condition of β-adrenergic stimulation or in the slowing of heart rate afterward once β-adrenergic stimulation ceases. Adjustment of the computational model parameters based upon recent experiments explore the functional changes caused by the CASQ2 mutation. In the simulation studies under rapid pacing (6 Hz), electromechanically concordant cellular alternans appeared under β-adrenergic stimulation in the CPVT mutant but not in the wild-type nor in the non-β-stimulated mutant. Similarly, the simulations of accelerating pacing from slow to rapid and back to the slow pacing did not display alternans but did generate early afterdepolarizations (EADs) during the period of second slow pacing subsequent acceleration of rapid pacing.

## 1. Introduction

Catecholaminergic polymorphic ventricular tachycardia (CPVT) is a cardiac arrhythmia induced by physical activities, emotional stress, or catecholamine infusion, which may further deteriorate into ventricular fibrillation (VF) [1]. The presence of β-adrenergic stimulation, which is common to all these triggering stimuli, increase L-type Ca^2+^ current, Ca^2+^ sequestration into the sarcoplasmic reticulum (SR) through phospholamban, and Ca^2+^ release from the SR via the ryanodine receptor (RyR2) through phosphorylation. The net result in an increase in myocyte Ca^2+^ content [2]. In fact, non-selective β-blockers are used as the standard treatment for CPVT [3,4]. The heart of CPVT patients does not display any morphological differences, and their pathogenicity is often not identified before the symptoms appear [5]. This malignant, young patient’s cardiac channelopathy has a 30–50% mortality rate [6]. CPVT type 2 (CPVT2) transpires by single nucleotide polymorphism (SNP) deletions or in the gene to express the Calsequestrin type 2 (CASQ2) protein [7,8]. It is an inherited autosomal recessive (both copies of allele mutated) trait of mutation. The CASQ2 expressing gene has 11 exons and encodes a protein containing 399 amino acids [9].

CASQ2 is a high-capacity and low-affinity Ca^2+^ buffering protein co-localized with the RyR2 channel in the lumen of the junctional SR. The binding site is the aspartic acid (Asp) rich region in the C-terminus where CASQ2 monomers aggregate to form dimers, then tetramers which turn into negatively charged Ca^2+^ binding pockets [10,11]. An extended C-terminal end comprises more than 70% acidic residues [12]. The three domains in the CASQ2 molecule consists of the amino-terminal loop, three thioredoxin domains, and a disordered acidic tail [13].

A total of fifteen CASQ2 mutations have been identified in humans, two out of them belong to deletion mutations—CASQ2^L23fs+14X^ and CASQ2^G112+5X^ [8]. Experimental studies revealed this mutation brings morphological changes in the SR by reducing the buffering capacity of CASQ2 in the SR luminal region and an increase in the volume of SR. The model incorporates these changes to mimic the SR changes by the onset of CASQ2^G112+5X^ mutant CASQ2 to study the mechanism of arrhythmia during β-adrenergic stimulation.

CASQ2^G112+5X^ is a homozygous deletion mutation and causes disruption of CASQ2 polymerization in protein expression. The deletion of 16 base pairs (c.339–354) causes a frameshift to generate premature stop codon after the removal of 5 amino acids. This mutation causes the omission of the entire II and III domains and parts of the first domain. The mutant protein lacks the total acidic residue required for the binding of Ca^2+^. The mutant also lacks the amino acids which are part of front-to-front or back-to-back interaction for the CASQ2 polymerization [14]. This mutation causes a reduction in the SR Ca^2+^ buffer, prolonged release of SR Ca+, an increase in SR volume, and impaired clustering of CASQ2 [15].

Computational models allow mechanistic observations and insights that are difficult or beyond observation in experimental studies. In the simulations, modification of the parameters for SR volume, CASQ2 buffer, L-type channel activities, and SERC2A pump activities in this newly developed stochastic Guinea pig myocyte model simulates CPVT2 and β-adrenergic stimulation in a Guinea pig ventricular myocyte carrying the CASQ2^G112+5X^ mutation. The simulation studies explore the mechanism of observed arrhythmia in patients. The present study demonstrated that alternans is the mechanisms responsible for generating arrhythmia during β-adrenergic stimulation in the mutant myocyte. The simulations also suggested that Ca^2+^ overload leading to increased RyR2 Ca^2+^ release and stimulation of the Na^+^-Ca^2+^ exchanger produced EADs when a rapidly paced myocyte returns to a resting pacing after catecholamine treatment. These results demonstrate how different pacing dynamics can lead to distinct types of arrhythmias in ventricular myocytes with the same genetic defect.

## 2. Materials and Methods

This is a computational study using a whole-cell stochastic computational model of excitation–contraction coupling in Guinea pig ventricular myocyte. There were no laboratory experiments using animals or isolated cells.

### 2.1. Model Development

The novel whole-cell stochastic computational model of excitation–contraction coupling in Guinea pig ventricular myocyte integrates a modified model of stochastic Ca^2+^ dynamics from the published rat model formulated by Williams et al. [16] with the published common pool model by Jafri et al. [17] for the Guinea pig ventricular myocyte [18]. The resulting model is the local control, Monte Carlo simulation model, with 20,000 stochastically gating Ca^2+^ release units that open in dyadic subspaces. The CRUs are cluster of 14 L-type and 49 RyR2 channels coupled with a dyadic subspace. Improving upon the previous rat ventricular myocyte models, this model integrates RyR2 adaptation to the gating mechanism of the intracellular Ca^2+^ [19,20,21,22]. The ionic current formulations of the new model are adapted from L-R models [23,24,25].

### 2.2. RyR2 Model

Jafri et al. [17] created a new model in 1998 by combining the L-R II [25] model with a more biophysically accurate formulation of myocyte Ca^2+^ dynamics by replacing the Ca^2+^ SR release mechanism in Luo-Rudy II with a dynamics RyR2 model with adaptation interacting the L-type Ca^2+^ channels in the dyadic subspace. There were four states in the RyR2 model: two closed states and two open states. Similar to the approach used previously, this model combined features of the previous model with the stochastic spark model [16,26]. The result is a new three-state model with two closed states and one open state [18]. As shown in Figure 1, the output is a new three-state model with two closed states and one open state. Figure 1 presents the second closed state (C_3_), which is an adaptive state. This RyR2 model’s gating mechanism formulations employ Monte Carlo simulation. Most RyR2s are in the closed state (C_1_) during the resting phase; however, when Ca^2+^ enters the dyadic subspace, the channels activate into an open state (O_2_) and followed by the channels transition into an adaptive state (C_3_).

In the new RyR2 model, the luminal regulation function (Φ) is determined by its dependency on Luminal Ca^2+^ regulation coefficient $\Phi_{m}$, and $\Phi_{b}$
(1)$$\Phi =\Phi_{m}\;{\left[Ca^{2+}\right]}_{sr}+\Phi_{b}$$
where ${\left[Ca^{2+}\right]}_{sr}$ represents both junctional and network Ca^2+^.

### 2.3. L-type Ca^2+^ Channel Model

The model includes a 6 state L-type Ca^2+^ channel model as shown in Figure 2. In this model state 2 (O_2_) and state 3 (O_3_) are open states, state 1 (C_1_) and state 6 (C_6_) are the closed states. The remaining two states (C_4_ and C_5_) are inactivated states. The inactivation of opening states of the L-type Ca^2+^ channel model happens in two separate ways—voltage-dependent inactivation, O_2_ → C_5_, and Ca^2+^ -dependent inactivation, O_2_ → C_4_. The Ca^2+^ in subspace is the one to controls inactivation in each release site. As the level of Ca^2+^ elevates in the subspace, it increases the rate of inactivation of L-type Ca^2+^ channels and prevents Ca^2+^ overload in the myoplasm [26]. The 6th state, C_6_ was added in the 5-state original model of Sun et al. [27] to have the stronger depolarization (≥−30 to ≤−40 mV) and all the channels during the resting period stay in this state. The Ca^2+^-dependent inactivation and voltage-dependent inactivation behaviors were updated from Morotti et al. [28].

During resting potential, most L-type channels are in a closed state (C_1_) and depolarization of the membrane potential facilitates their transition into an open state (O_2_). Channel in O_2_ state may continue to open state (O_3_), transition into the inactivated state (C_5_), or may transition into another inactivated state (C_4_) through a Ca^2+^-dependent process. Details of the model are available in our previously published article [18].

Modifying the parameters related to β-adrenergic stimulations and CASQ2^G112+5X^ mutations in the model simulated experimental changes that explained and enabled comparison of mutant myocyte behavior and morphological features to wild-type myocyte behavior. This paper examines the impact of the mutation on Ca^2+^ dynamics in intracellular compartments during normal and rapid pacing compared under control and β-adrenergic stimulation in both wild type and mutant myocytes to see if any arrhythmia occurs and, if so, what mechanisms are involved. Modification of the parameters for L-type Ca^2+^ channels and SERCA2-ATPase pump cycling rates (V_cycle_) allows model simulation of β-adrenergic stimulation consistent with experiments. Likewise, for the CASQ2 mutation, the luminal Ca^2+^ regulation coefficient, ($\Phi_{m})$ of RyR2 was increased and two morphological features (SR volume increase and reduction in CASQ2 buffer) were added. Terentyev et al. [29] in their experiment in rat ventricular myocyte reported that there is a 2-3-fold increase in Ca^2+^ sparks due to an increase in the volume of luminal Ca^2+^. Similarly, Kornyeyev et al. [30] reported that the loss of CASQ2-mediated Ca^2+^ buffering caused a faster rise in luminal free Ca^2+^. Song et al. [31] reported an enhanced SERCA2a pump SERCA-ATPase Cycling rate (V_cycle_) activities in CASQ2 deficient myocytes. CASQ2^G112+5X^ mutation behaves similarly to knockout CASQ2 (CASQ2^−/−^) [15] and Knollmann et al. [32] found ~50% SR volume was increased in CASQ2^−/−^ deficient ventricular myocyte to compensate the total loss of Ca^2+^ buffer.

### 2.4. Simulation Protocols

First, simulation of wild type and mutant myocytes at 1 Hz pacing achieved qualitative agreement with steady-state Ca^2+^ dynamics in each type of myocytes. For further simulations, these conditions served as the initial state for our model. The simulations performed at 1 to 6 Hz pacing include the following: (a) wild-type control pacing, (b) mutant myocyte control pacing, (c) wild type β-AR stimulation pacing, and (d) mutant myocyte -adrenergic stimulation pacing. Table 1 shows the updated simulation parameters for β -AR stimulation in wild type and mutant myocytes. β-adrenergic receptor stimulation increases SR Ca^2+^ content by increasing L-type current and SERCA2a activity [33]. The following features represent the β-adrenergic stimulation in a WT and CASQ2 mutant myocytes. Figure 3 shows that the CPVT mutation in simulated in Guinea pig ventricular myocytes shows similar behavior to those observed in experiments in mice [32].

#### β-AR Stimulation Parameters

During β-AR stimulation the following parameter changes were made

(a)Increase in L-type channel activity: Experiments observe an amplified L-type current when the level of catecholamine increases by the activation of adrenergic receptors in the sarcolemma. An increase of 40% for L-type current (P_dhpr, the single channel permeability) replicates experimental findings. Ginsburg and Bers [33] found a 53% increase in the L-type Ca^2+^ channel peak value with the treatment of isoproterenol (ISO).(b)Increase in luminal dependence: CASQ2 is the major Ca^2+^ storage protein in the SR and its absence has a major implication in the availability of free Ca^2+^ in the SR luminal. Luminal sensitivity regulation function (**ϕ**) depends upon free Ca^2+^ in the SR as shown in Figure 1. Greater free Ca^2+^ available in the lumen enhances the opening rate of RyR2. Since there is a total reduction in Ca^2+^ buffer in CASQ2^G112+5X^, a 90% increase in luminal dependence (K_jsro_) provided a steady-state Ca^2+^ transient by distressing free SR Ca^2+^ availability. Gyorke [35] reported the RyR2 open probability increased 0.26 ± 0.04 to 0.49 ± 0.09 with doubling luminal doubling SR free [Ca^2+^].(c)Enhanced SERCA-ATPase cycling rate (V_cycle_): Increased Ca^2+^ in cytosol due to an increase in L-type current as well as increased RyR2 channel activity, affects the SERCA2A cycling rates, and they pump Ca^2+^ back to SR rapidly increasing SR load [16,36,37]. In the simulation, the whole-cell SERCA pump flux is given by
(2)$$J_{serca}=2v_{cycle}\;A_{p}$$
where $v_{cycle}$ is cycling rate per molecule, $A_{p}$ is the concentration of SERCA molecules per liter cytosol and the unit of Ca^2+^ flux, and $J_{serca}$ has units of mol s^−1^.(d)Increased SR volume: The morphometric analysis of volume fraction of SR membrane of CASQ2 knockout myocyte by Knollmann et al. found that the SR to cytoplasm volume ratio increased by ~51% while the volume related to myofibril was increased by ~45% [32]. They also found a 50% increase in the SR volume which effectively maintains the normal SR Ca^2+^ storage capacity.(e)Abolition of SR buffering capacity: CASQ2^G115+5x^ deletion mutation removes the entirety of CASQ2 protein leaving no place to SR Ca^2+^ to buffer [15]. The amount of SR Ca^2+^ was reduced to 5% of control to reflect residual Ca^2+^ buffering in the SR.

### 2.5. Numerical Methods

The program to solve the differential equations of the model used the PGI CUDA Fortran (Fortran95) compiler to execute the program on an Ubuntu Linux platform (12.04). CUDA (compute unified device architecture) is a parallel computing platform and programming language developed for graphic processing units (GPUs) by NVIDIA. Calcium dynamics at a single-channel level used the Ultra-Fast Markov chain Monte Carlo (UMCMC) method for the stochastics gating from CRUs [38]. The graphs and data analysis used IDL software and the Python programming language. Numerical solution of the systems ordinary differential equations employed the explicit Euler method. The time step is for the differential calculation is 10 ns.

### 2.6. Statistical Analysis

To ensure that the simulations were consistent each simulation was run 10 times with different starting seed values for the pseudo random number generator. Simulations were compared to make sure that the results were similar across simulations. A representative simulation was chosen to display in the figures. Microsoft Excel was used to calculate the means for these properties and the two-sided Student’s *t*-test comparing the means under the assumption of different variances and sample sizes.

### 2.7. Model Simulation Protocols

The normal heartbeat of an adult Guinea pig is ~240 beats per minute (4 Hz). The heartbeat rises above 5–6 beats per second (5–6 Hz) when β-adrenergic receptors are stimulated by exercise or emotion. Below 4 Hz pacing, simulations of both WT and mutant myocytes produces stable beat-to-beat transients of all ionic currents and action potential measured. Action potential anomalies in the form of alternans were identified when mutant myocytes were paced at 6 Hz with β-adrenergic stimulation. Table 2 shows the simulation protocols and parameter changes in both WT myocyte and mutant myocyte in different pacing frequencies. In addition to alternans simulations, slow–rapid–slow simulations with 1-5-1 Hz pacing using the same protocols and parameters as the β-adrenergic stimulation were performed. The purpose of this simulation was to monitor EADs, and it was run for 30:10 s of slow pacing, 10 s of rapid pacing, and 10 s of slow pacing. Each simulation was repeated 10 times with different seed values for the pseudo random number generator to make sure that the observations were repeatable. With the parameters used in this study, similar behavior was reproduced every time. A representative simulation is shown in the figures presented.

## 3. Results

These studies simulated myocytes from both wild-type and CASQ mutant myocytes. β-adrenergic activation of the CASQ2 mutant myocyte in our model revealed three CPVT-related phenomena: (1) EADs, found at low frequency immediately after switching from rapid pacing, (2) alternans, during rapid pacing, and (3) alternative beat skipping, also observed in rapid pacing. This is a crucial feature of this model: it simulated these arrhythmia events without changing any parameters other than the pacing frequency.

### 3.1. Simulation at 1 Hz

Our first set of simulations explored how well the model reproduced the Ca^2+^ dynamics observed in experiments of a mouse model of CPVT. The four simulation protocols specified in Table 2 were applied to model simulations at 1 Hz. The 1 Hz simulation was run for 10 s and the last five Ca^2+^ transients were measured for their maximum values. The diastolic Ca^2+^ concentration was measured just prior to the beat. The values are shown in Table 3. When compared to the wild-type simulation the average Ca^2+^ transient maximum for the mutant was significantly different significantly (*p* < 0.05) from the mutant myocyte, the wild-type myocyte with β-adrenergic stimulation, and the mutant myocyte with β-adrenergic stimulation. Similarly, the diastolic [Ca^2+^] was significantly (*p* < 0.05) from the mutant myocyte, the wild-type myocyte with β-adrenergic stimulation, and the mutant myocyte with β-adrenergic stimulation. When compared to the experiments by Knollman and co-worked in transgenic mice [32] the simulation follow similar patterns of change.

### 3.2. Slow–Rapid–Slow (S-R-S) Pacing Developed EADs

In the experiments involving CASQ2 mutations associated with CPVT, EADs were detected at slow beating of the heart following periods of rapid pacing, possibly due to the prolonged APD with slow pacing and increased SR Ca^2+^ load following a period of rapid pacing. Therefore, the simulations studies presented here explore the mechanisms of EADs occurrence at lower pacing frequencies after rapid pacing by comparing it to the slow pacing simulation before the rapid pacing with the identical conditions.

The simulation to understand the mechanism behind this experimentally observed behavior uses features of the CASQ2 mutant, CASQ2^G112+5X^ following the protocols with slow, fast and-slow phases described above. The simulated mutant myocyte was paced for 10 s at 1 Hz, followed by 10 s at 5 Hz, then for another 10 s at 1 Hz. The mutant displayed normal pacing in low frequency, and then paced well without any irregularities in 5 Hz before returning to slow pacing (1 Hz) where EADs were observed in numerous beats (Figure 3A). As shown in Figure 4A, EADs were observed in the action potential of 22, 23, 26, 27, 28 and 29 s during the second phase of slow pacing. In the WT myocyte, a similar simulation with β-adrenergic stimulation produced normal APs either before or after the rapid pacing (Figure 4B). During β-adrenergic stimulation, mutant myocytes had higher average APDs than WT myocytes (229.1 ± 15.93 vs. 211.7 ± 10.1). Quantification of simulated Ca^2+^ sparks for the mutant in each beat before and after rapid pacing showed that the second phase had more than double the number of sparks than the first phase (155,082 ± 8638 vs. 368,841 ± 26,994) (Figure 4C). A two-tailed student’s *t*-test unequal variance finds that the mean number of sparks are unequal in panels (C) and (D) with *p* < 0.001.

Plotting the data and testing for statistically significant changes in the mean values of the plotted data helps identify differences in Ca^2+^ dynamics. The key differences between pacing at slow pacing rate before the period of rapid pacing in the mutant myocyte is the higher concentration of SR Ca^2+^ after the rapid pacing as demonstrated in the wild-type myocyte under β-adrenergic stimulation (Figure 5A and Table 4) with *p* < 0.001 using a two-tailed student’s *t*-test unequal variance. Furthermore, a two-tailed Student’s *t*-test unequal variance finds that the mean [Ca^2+^] before and after rapid pacing in mutant myocytes are unequal with *p* < 0.001. This allows the second slow phase to generate more Ca^2+^ sparks than the first one (Table 4) consistent with the finding of Guo et al. [39] who reported the Ca^2+^ spark frequency increases with increased SR load.

The EADs that occur in the elongated APDs (Figure 5B) are display a statistically higher mean value (*p* < 0.001) and higher variability in the average APD in the second slow phase (330 ± 56.95) than in the first slow phase (229.1 ± 15.93 where the APs in the second phase did not have EADs. The mutant myocyte with β-adrenergic stimulation before rapid pacing and the wild-type myocyte with β-adrenergic stimulation have means significantly higher APD (*p* < 0.001) than wild-type myocytes under control conditions.

Increased electrogenic I_ncx_ (Figure 5C) due to elevated cytosolic calcium also supported further elongation of the APD. There are statistically different means for the mutant myocytes with β-adrenergic stimulation before and after rapid pacing (*p* < 0.001). The mutant myocyte with β-adrenergic stimulation before rapid pacing and the wild-type myocyte with β-adrenergic stimulation have means significantly higher I_ncx_ (*p* < 0.001) than wild-type myocytes under control conditions. Interestingly, the mutant myocyte with β-adrenergic stimulation before rapid pacing displays significantly (*p* < 0.001) higher mean I_ncx_ than the wild-type myocyte with β-adrenergic stimulation. It has been suggested that the generation of the EADs is related to the elongated action potential and late I_Ca_, late I_Na,_ or inward I_ncx_ currents (Figure 5C) [40,41]. Slightly higher action potential amplitudes (39.3 ± 0.33) are noticeable in the second slow phase over the first one (38.47 ± 0.22), but their values were not spread out enough to support idea that the amplitudes may play any role in the generation of EADs.

The higher average Ca^2+^ sparks count (Figure 5D) per beat after rapid pacing (247,320 ± 68,967) than before rapid pacing (123,847 ± 8638) with the same pacing frequency should have played a major role in destabilizing the myocyte (*p* < 0.001). Further analysis showed that the first slow phase had 155,082 ± 8633 Ca^2+^ sparks per second compared to 368,841 ± 26,995 in the second slow phase after the period of rapid pacing (Table 4). The mutant myocyte with β-adrenergic stimulation has a higher Ca^2+^ spark amplitude with wild-type either with or without β-adrenergic stimulation (*p* < 0.001).

Further evidence of the changes in Ca^2+^ dynamics is displayed at the Ca^2+^ spark level. In the second slow phase, the difference between per sec Ca^2+^ sparks per beat is four times bigger than in the first slow phase (122,521 ± 12,252 vs. 31,235 ± 3452) as shown in Table 5. The Ca^2+^ spark frequency per second in the WT myocyte was 60,495 ± 2439, whereas the Ca^2+^ spark frequency per beat was 50,694 ± 1986. The average spark amplitudes, on the other hand, showed a different pattern, being higher in the first sluggish phase (64.27 ± 0.52) than the second (62.07 ± 0.51) (Table 5).

### 3.3. Mechanism of EADs

To better understand the mechanism behind experimentally observed EADs, simulations with the mutant myocytes that displayed EADs during slow pacing that follows the period of rapid pacing were studied (Figure 4. Comparison of multiple graphs based on these simulations shown in Figure 6 demonstrate how the EADs develop. The action potentials before are normal and the action potential after rapid pacing shows and EAD (Figure 5A, blue and red, respectively). Figure 6B shows that the EAD is accompanied by increased L-type Ca^2+^ current. After rapid pacing, a larger average number of Ca^2+^ sparks are present (Table 4), that result in increased RyR2 open probability (Figure 6C). Analysis of the simulations results explain the sequence of events leading to the EAD. The late reactivation of L-type Ca^2+^ channel (Figure 6B) occurred approximately 0.421 s (in the first action potential in Figure 6B), while the reopening of RyR2 (Figure 6C) occurred around 0.405 s, indicating that spontaneous RyR2 opening occurred before L-type Ca^2+^ channel reactivation. The shift of the curve to the right in [Ca^2+^]_nsr_ (Figure 5D) occurred at 0.407 s because of spontaneous Ca^2+^ release. A significant variation in the Na^+^-Ca^2+^ exchange current occurs with the availability of extra Ca^2+^ in the cytoplasm when comparing the first and second phases (Figure 6E). When Ca^2+^ was extruded from the myocyte, the I_ncx_ generates an inward current causing depolarization of the membrane potential. The I_ncx_ amplitude increases from 0.587 to −0.856 in the first phase of slow pacing, as shown in Figure 6F, with an absolute rise of 1.443 (μA/cm^2^). During the second phase of slow pacing, the I_ncx_ current increases by 1.752 (μA/cm^2^), stretching from 0.577 to −1.172. In the second slow phase, it is 0.309 (μA/cm^2^) higher than in the first slow phase. Furthermore, it extended from 0.48 to −1.55 at 5 Hz rapid pacing, with an absolute rise of 2.03 (μA/cm^2^).

### 3.4. Action Potential and Ionic Currents during Rapid Pacing

Although the SR Ca^2+^ overload caused by rapid pacing triggered spontaneous Ca^2+^ release in slow pacing that followed, the action potentials and other ionic currents were found to be normal during 5 Hz pacing. Analysis of the simulated action potential and other Ca^2+^ time courses between 18 and 19 s showed no irregularities in those components during the time of rapid pacing (Figure 7). The action potential plot (Figure 7A) showed no abnormality, whereas the L-type Ca^2+^ channel current plot (Figure 7B) showed a normal reduction in amplitude with increasing rate but no anomaly. Similarly, the I_ncx_ (Figure 7C) results were normal, with no unusual increases in amplitude. The RyR2 channels were late reactivated in the plots for the Ca^2+^ transient [Ca^2+^]_myo_ (Figure 7D), SR Ca^2+^ release [Ca^2+]^_nsr_ (Figure 7E), and RyR2 Po (Figure 7F), but the release was not enough to cause changes in the AP.

### 3.5. Alternans and Alternately Skipping Beats

The next set of simulation studies explored the mechanism behind alternans during rapid pacing. In the pacing frequency range of 1 Hz to 6 Hz, the myocytes simulated for control, mutant control, and WT β-adrenergic receptors were mostly normal. Overall, the APD in β-adrenergic stimulated myocytes was longer than in WT or control mutants. The missing beats observed in a longer simulation, alternans and the missing beats appear in Figure 8. Alternans were abundant, before or after beat skipping but the alternate beat skipping phenomenon occurred between 18.5–20.5 s. As expected, SR Ca^2+^ load was larger in β-adrenergic stimulation myocytes than in control myocytes, and it increased as the pacing frequency increased. When comparing control and mutant myocytes, mutant myocytes had a lower SR Ca^2+^ level due to higher opening probability of RyR2. The mutant myocyte activated with β-adrenergic simulation beat normally until 5 Hz pacing, but it exhibited abnormalities in the action potential at 6 Hz pacing as shown in Figure 8. Alternans with the alternate beat missing were visible in a 30-s simulation (Figure 9). The alternans are both mechanical (retrenchment amplitude) and action potential duration (APD or electrical) alternans. From 18.5 to 20.5 s (Figure 9), the alternate beat was missing for about 2 s, with alternans of various shapes and sizes noted the rest of the time. The simulated action potential plot had similar patterns of alternate action potentials as the experimental data (Figure 9) [42].

### 3.6. Mechanism of Alternans

Concordant mechanical and electrical alternans were observed during 6 Hz pacing. Dissecting the 6 beats between 8.5 and 9.5 s demonstrated the mechanism behind the action potential alternans shown in Figure 10A. As indicated in Table 5, WT myocytes had an average APD of 141 ± 3, which is higher than shorter beats but lower than longer beats in alternans (Table 4). The average peak action potential amplitude in WT 38.38 ± 0.4 is higher in both shorter and longer beats in alternans when comparing peak action potential amplitudes. The durations, amplitudes, and standard deviations revealed that the mutant myocyte had both mechanical and electrical alternans. The activities of Ca^2+^ sparks in each subsequent beat (Table 5) were analyzed by counting them (Figure 9B) and determining their average and peak amplitudes each beat. In comparison to control myocytes, both shorter and longer beats had a large amount of Ca^2+^ sparks. The longer beat displayed one-third higher sparks than the shorter beat (Table 5 and Figure 10B). Sparks had shorter average amplitudes (57.03 ± 0.01 vs. 60.99 ± 0.02) and peak amplitudes (180.11 ± 0.03 vs. 190.79 ± 0.03) than wild type myocytes. A two-tailed Student’s *t*-test with unequal variances finds that the means of the number of sparks in the long and short beats are unequal with *p* < 0.001. The measured time between two successive beats is the diastolic interval. The diastolic interval between shorter and longer beats (0.025 ms) was significantly greater than the diastolic interval between longer and shorter beats (0.001 ms) (Figure 10). A longer diastolic interval allowed the SR Ca^2+^ load for an incoming beat to be increased, and it was shown to be higher.

Other ionic currents and Ca^2+^ transients were also displayed between 8.5 and 9.5 s. The L-type current (Figure 10C) showed opposite alternating behavior to the respective APs, as well as L-type Ca^2+^ channel opening probability P_O,LCC_, and opening fraction of L-type Ca^2+^ channel states due the Ca^2+^-dependent inactivation (Figure 10D,E). During the systolic phase, the majority of Ca^2+^ channels undergo Ca^2+^-dependent inactivation (C_4_) and voltage-gated inactivation (C_5_), and finally reach to inactivation state (C_6_). When the SR load was larger, the open proportion of L-type Ca^2+^ channel states revealed that L-type Ca^2+^ channel inactivation was mainly Ca^2+^-dependent as shown in Figure 10E.

Pacing at 6 Hz pacing elevates the SR Ca^2+^ over lower pacing rates and displays alternans (Figure 10A). The larger reductions in [Ca^2+^]_nsr_ are accompanied by larger RyR2 release events as indicated by the increased RyR2 open probability (Figure 10B). Similarly, the higher peak diastolic [Ca^2+^]_nsr_ preceded the larger RyR2 open probability events due to the luminal dependence of RyR2 opening. The beat-to-beat variation in the SR Ca^2+^ release result in the variations in [Ca^2+^]_myo_ (Figure 11C) which in turn was responsible for the alternate electrogenic I_ncx_ current (Figure 11D). Those I_ncx_ currents were alternate but unlike L-type currents, their alternating pattern was aligned to the APs, longer the APs, longer the I_ncx_ and vice versa, and the electrogenic nature helped elongated APs alternately. The plots of I_Na_ (Figure 11E) and the Na^+^ channel inactivation gate (Figure 11F) followed the same pattern of APs beats in contrast to the alternate patterns displayed by the L-type channels. Although Na^+^ channels are responsible for the initial depolarization of the membrane potential that activates the L-type channels. The Ca^2+^ dependent inactivation caused a smaller L-type current to occur in the same beat with larger action potential, RyR2 Ca^2+^ release and Na^+^ current. Table 6 shows the visual alternate patterns of ionic currents, action potential and SR Ca^2+^ transients in consecutive beats. Alternating Na^+^ current causes the alternating amplitude of the action potential, e.g., full activation of Na^+^ channels generated taller action potentials, as well as the alteration in I_ncx_. Incomplete recovery of Na^+^ channels from the inactivation in the previous beat causes the alteration in I_Na_. Furthermore, during APD alternans the diastolic interval becomes shorter which means shorter time for Na^+^ channels to recover from inactivation this makes a smaller number of channels are available for incoming beat hence APD alternans also plays role in amplitude alternans. In the meantime, the I_ncx_ gets elongated with the Ca^2+^ availability in the cytoplasm and assists the APD to increase further.

### 3.7. Nonrecovery of Sodium Channels Results into Alternate Beat Skipping

Figure 12A displays longer action potentials with few skipped beats in between 18.5 to 20.5-s segment. The L-type Ca^2+^ current was also missing (Figure 12B) concordant with action potentials in contrast to the alternans observed earlier. The I_Na_ currents failed to activate with the skipped beasts (Figure 12C) indicating there was no triggering of the action potential. In contrast during alternans, the Na^+^ channels (Figure 10E) were partially activated, and even very low activation was able to activate enough L-type Ca^2+^ channels and an action potential. The plots of Na^+^ inactivation gate (Figure 12D) displayed all the Na^+^ channels did not recover from the previous inactivation even late in the systolic phase. Na^+^-Ca^2+^ exchange current (Figure 12E) and NSR Ca^2+^ level (Figure 12F) was gradually decreased during the beat-missing period and recovered when the alternate beats started.

## 4. Discussion

CASQ2 mutations disrupt intracellular Ca^2+^ dynamics, resulting in CPVT in the heart. Researchers have proposed different mechanisms in both experimental and simulated contexts. A series of simulations in WT and mutant myocytes under adrenergic stimulation tested the hypothesis about these processes, and the results are shown above. CPVT is a hereditary malignant arrhythmia that can manifest itself in people with no prior symptoms or diagnosis, leading to abrupt cardiac death. This work adds mechanistic evidence to the experimental observation that excess diastolic SR Ca^2+^ is linked with leaky RyR2, resulting in DADs. CASQ2 controls RyR2 as a luminal Ca^2+^ sensor, according to Gyroke et al. [43], while Knollmann et al. [32] showed that in a null CASQ2 myocyte, RyR2 can sense luminal Ca^2+^ and manage intracellular Ca^2+^ normally in low SR Ca^2+^, but this may change in higher SR Ca^2+^ load. The model was able to capture this idea, with the increased SR Ca^2+^ load producing alternans in the continuous rapid pacing but with the occurrence of spontaneous Ca^2+^ release if the rapidly pacing myocyte passed in low pacing or pause after and the EADs are produced as the precursor of CPVT.

The arrhythmias associated with catecholaminergic polymorphic ventricular tachycardia (CPVT) is induced by β-adrenergic stimulation that accompanies physical activities, emotional stress, or catecholamine infusion, which may further deteriorate into ventricular fibrillation (VF) [1]. β-adrenergic receptor stimulation activates the associated G protein. The increases 3′-5′-cyclic adenosine monophosphate (cAMP) production which activated protein kinase A (PKA). PKA phosphorylates many intracellular targets including the L-type Ca^2+^ channels, slow delayed rectifier potassium channels, fast sodium current channels, the cystic fibrosis transmembrane conductance regulator, the sodium–potassium pump, the ryanodine receptors, phospholamban, troponin I, myosin binding protein-C, and titin [2]. The modeling study presented here studies the targets of PKA that are most involved in altering Ca^2+^ dynamics in the cardiac myocyte–increasing L-type Ca^2+^ current, RyR2 open probability, and SERCA activity through phospholamban.

### 4.1. Alternans

The mutant CASQ2 mouse ventricular myocytes simulations at 6 Hz pacing produce Ca^2+^ alternans. The simulations suggest that cellular alternans develops under β-adrenergic stimulation in the CPVT mutant, but not in the wild-type and unstimulated mutant under rapid pacing (6 Hz). Action potential (APD and amplitude), ionic currents, and Ca^2+^ transients all changed from beat to beat. In experiments, it is difficult to pinpoint the responsible ionic behavior that produces alternans at the cellular level because of this multi-faceted impact on AP. The advantage of multi-scale modeling is that it allows us to trace out and detect each component’ ‘s real-time relationship with the action potential. The following major points are drawn from the model:I_Na_ plays major role in the action potential amplitude alternans and SR Ca^2+^ has the main role in the APD alternans.Na^+^ channel alternation has negative feedback on the l-type current alternation.Intracellular Ca^2+^ has negative feedback to extracellular Ca^2+^ in SR overload.Very short diastolic interval for the refilling of Ca^2+^ after a beat with longer APD.Higher fractions of L-type channels exhibit Ca^2+^-dependent inactivation and voltage-dependent inactivation.Increased NCX current (I_ncx_) increases the APD.

In rapid pacing, the Na^+^ channels are unable to recover from previous beat inactivation and they produce shorter Na^+^ current in the incoming beat. The reduced amplitude of I_Na_ would activate fewer L-type Ca^2+^ channels reducing the amount of Ca^2+^ influx. However, this loss of excitation–contraction coupling would be compensated for by the activation of adrenergic receptors which increases L-type Ca^2+^ current and RyR2 Ca^2+^ sensitivity. The simulations showed that a larger I_Na_ ended up having a smaller L-type Ca^2+^ channel current. The level of [Ca^2+^]_SR_ controls this negative feedback of I_Ca_ towards I_Na_. The smaller release of Ca^2+^ in shorter beat leaves higher residue in the SR in the first place and subsequent SR refilling occurs on top of an existing residue. The shorter beat accompanies longer diastolic interval which allows a longer time for Ca^2+^ reuptake to SR. All these events create Ca^2+^ overload in the SR for an upcoming beat. When RyR2 channels are activated by Ca^2+^ via the L-type Ca^2+^ current, the Ca^2+^ release from RyR2s in the SR potentiates dyadic subspace Ca^2+^ causing the L-type Ca^2+^ channels to undergo Ca^2+^-dependent inactivation to prevent overloading of intracellular Ca^2+^. However, when Ca^2+^ is elevated and the RyR2 opening is pathologically increase electrical and mechanical alternans occurs. Simultaneously, there also are increases I_ncx_ producing more depolarizing current. The intracellular Ca^2+^ released from SR is responsible for the generation of alternans in cardiac myocyte and the positive coupling of I_ncx_ further increases the action potential duration.

The spark analysis of the simulation result indicated that more than one-third of Ca^2+^ sparks occurred in the longer beats as opposed to the shorter ones governed by fluctuation in Ca^2+^ content in the SR. Diaz et al. [44], in their experiment, reported a measurable change in the SR Ca^2+^ content during alternans. They also found reduced openings of L-type Ca^2+^ channels during alternans, similar to patterns in L-type Ca^2+^ channels due to Ca^2+^-dependent inactivation observed in the model.

L-type channels have two types of inactivation states–voltage-dependent inactivation and Ca^2+^-dependent inactivation to prevent the pathological overload of Ca^2+^ during prolonged depolarization. Primarily elevated intracellular Ca^2+^ concentration near the junction of SR (dyadic subspace) triggers channel inactivation providing negative feedback to Ca^2+^ influx. The inactivation rate is very high when there is very high [Ca^2+^]_ds_ in dyad in comparison to bulk myoplasm. Kubalova [45] distinguished two phases of Ca^2+^ inactivation of L-type channels—a slow phase that depends on Ca^2+^ flow through the channels (Ca^2+^ current-dependent inactivation) and a fast one that depends on Ca^2+^ released from the SR, Ca^2+^ (Ca^2+^ release-dependent inactivation). Hence, SR released Ca^2+^ is the most effective inactivation mechanism in the inhibition of Ca^2+^ entry through the channel. The inactivation of the L-type Ca^2+^ channel shown to depend linearly on the rate and magnitude of the Ca^2+^ release from the RyRs [46]. Experiments also have suggested that the SR Ca^2+^ serves as the feedback mechanism to the L-type channels and their amplitude decreases with higher SR Ca^2+^ release [45].

Our finding agrees with Saitoh et al. [47], who demonstrated in dog’s ventricular myocyte experiment that the APD alternans are controlled by intracellular Ca^2+^. The end-systolic SR volume is increased after a shorter beat which leads to a greater end-diastolic volume for the next longer beat and this process is more prominent in the rapidly paced heart [48]. The APD depends upon the preceding diastolic interval, where the longer the diastolic interval the greater the APD, and vice versa [49]. In this model, the longer diastolic interval precedes a longer action potential and short diastolic interval is followed by shorter action potential. Previous studies suggested that I_ncx_ is responsible for the prolongation of APD during large Ca^+^ transient [50]. Since three Na^+^ ions enter the cell for every Ca^2+^ ion extruded, this increase in driving force elevates the inward membrane current which prolongs the APD.

In the pattern of the longer and shorter beat in alternans, sometimes longer beat is prolonged and the shorter beat goes missing. This is also a form of alternans, but the model suggests different mechanisms other than SR Ca^2+^ overloading in this case. The alternate beat missing is caused by incomplete recovery of Na^+^ channels from inactivation (h-gate) during the relative refractory period [51,52]. The relative refractory period is the period in between action potential depolarization and enough number of Na^+^ channels are available to initiate the incoming beat. Within the milliseconds of their activation, most of the Na^+^ channels undergo inactivate which is faster than total deactivation. The inactivated channels gradually reach the closed state and by the time the relative refractory period arrives, the depolarizing stimulus has passed. In rapid pacing, Na^+^ channels have the relative refractory period where they have not fully recovered from the previous inactivation. During alternans, simulations show a partially inactivated Na^+^ current as well as alternate action potentials and an increased I_ncx_. The model suggests that the longer the plateau, the larger the I_ncx_, which increases the late influx of Na^+^ current consistent with experiment [53]. When the Na^+^ channels are still inactivated at the time of new beat Na^+^ channels fail to activate to depolarize the membrane and the voltage-gated L-type Ca^2+^ channels fail to activate resulting in a skipped beat.

Many researchers believe there are two ways DADs can occur during β-adrenergic stimulation, the aberrant leaking of SR Ca^2+^ during diastole and/or excessive Ca^2+^ influx via L-type channel [54]. Observations in the model indicate that the SR was able to manage Ca^2+^ overload by increasing RyR2 opening probability allowing for more Ca^2+^ release from SR. The model simulations also indicated that under these conditions more L-type channels undergo Ca^2+^-dependent inactivation after a longer beat. After a long beat, there is a very short diastolic phase and might not have enough time required for reloading of SR for the upcoming beat which makes the beat a shorter beat. Although simulations had 40% L-type increases during β-adrenergic stimulation, a diastolic activation of L-type channels and the ensuing Ca^2+^ influx was never observed. Based on simulation findings, these studies are unable to support the DADs being the mechanism of an arrhythmia in the heart having mutation in the protein of CASQ2 expressing genes under the rapid pacing seen during β-adrenergic stimulation.

### 4.2. Early Afterdepolarization (EAD)

There are three mechanisms of EADs—spontaneous SR Ca^2+^ release caused by intracellular Ca^2+^ loading, β-adrenergic stimulation during stress or exercise, and a resurgence of electrogenic NCX current [55,56]. However, there is no agreement in the interrelation among those mechanisms. Weiss et al. [56] reported that EADs occurred due to the reduction in the repolarization reserve but in the simulation an increase in depolarization reserve is responsible for generating EADs. They also claimed EADs occur in a heart during bradycardia, but our study finds EADs emerge when a rapidly pacing myocyte enters slow pacing mode. Many researchers agree that APD prolongation is necessary to aid EADs, which was also suggested by the model. In the simulations, the irregular intracellular Ca^2+^ dynamics caused by β-adrenergic receptors in the mutant myocyte caused elongated APD. Weiss et al. [56] suggested that the primary major current to produce EAD is I_Ca_ and the second major current is I_ncx_ but in the simulations, the primary reason causing EADs is spontaneous Ca^2+^ release and the secondary causes are I_ncx_ and I_Ca_. Iyer et al. [57] stated that the stabilization of the SR Ca^2+^ is important in reducing the probability of spontaneous Ca^2+^ release but the stabilization factor, CASQ2 is deleted in CASQ^G112+5x^ mutation and it provoked all the instability in the mutant myocyte. January et al. and Antoons et al. suggested that EADs were caused as a result of reactivation of L-type channels following the prolonged plateau phase however, in the simulations spontaneous Ca^2+^ release occurs before reactivation of L-type channels [58,59]. Because of adrenergic stimulation during exercise or stress, the L-type channels continuously releasing Ca^2+^ into the dyadic subspace even late in the plateau phase and the same phenomenon is observed in the simulations also but analysis of the simulations could not support it as a primary source of EADs. At the same time, it was seen the electrogenic I_ncx_ current is also higher than the prior slow phase.

These simulation results are supported by the experimental literature. In experiments, the APD lasted longer due the inward current upon repolarization [60,61]. Similarly, there is a variation in the level of SR Ca^2+^ before and after rapid pacing. Figure 6D shows that following rapid pacing, the diastolic Ca^2+^ level is higher (1092 M vs. 1009 M) than during the slow phase. According to Volders et al. [62], arrhythmogenic responses are followed by spontaneous Ca^2+^ release during systole, and inward I_ncx_ contributes to the production of EADs during β-adrenergic stimulation. The simulations suggests that spontaneous Ca^2+^ release from the overloaded SR and the elongated APD caused by electrogenic inward current I_ncx_ causes EADs to happen. This is consistent with the experimental finding by Priori et al. [63] that blocking NCX current by benzamil suppressed the EADs. In a rabbit model, by reducing 55% SR Ca^2+^ uptake of ISO exposed myocytes, Xie et al. [64] were able to control spontaneous Ca^2+^ release and EADs. Simulations accelerating pacing from slow (1 Hz) to rapid (5 Hz) and back to slow (1 Hz) did not display cellular alternans but did generate early afterdepolarizations (EADs) during the period of slow pacing that followed the period of rapid pacing. The model suggests that the rapid pacing loads the cytosol and sarcoplasmic reticulum with Ca^2+^, which in the CPVT mutant with increased RyR2 open probability can trigger spontaneous Ca^2+^ release which activates Na^+^-Ca^2+^ exchange resulting in action potential prolongation. These studies suggest that spontaneous Ca^2+^ release due to SR overload is the underlying cause of the arrhythmia in these patients.

There are potential study limitations. The model simulations would produce consistent results across the 10 simulations for each simulation protocol (numerical experiment). This is not unusual for computational studies because parameter variations are chosen that produce the normal and pathological behavior consistently to ease study of the underlying mechanisms. In CPVT in patients, the 4-year ad 8-year observed event rate is 31% and 58%, respectively [65], with a mortality rate of 31% by age 30 [65]. This might be heterogeneity of the cardiac ventricular myocyte. For example, the ion channel expression of the cardiac ventricular myocyte varies transversely across the heart wall as well as form apex to base [66,67]. There is likely to be variation as well from patient to patient. Future studies can explore the effect of the variability of the ion channel expression on the potential for generating arrhythmia.

### 4.3. Clinical Correlations

The simulation studies demonstrated the mechanisms of arrhythmia and the role of aberrant Ca^2+^ dynamics and membrane currents. The simulations show that Ca^2+^ overload is an important feature in producing the EAD’s seen with the reduction in pacing frequency after a period of rapid pacing. The phosphorylation of the L-type Ca^2+^ channel increases Ca^2+^ entry into the myocyte. The phosphorylation of phospholamban increases the sequestration of Ca^2+^ into the SR. The increase in RyR2 open probability by phosphorylation increases the number of spontaneous sparks (Ca^2+^ leak) that leads to triggering of the EADs. These defects align with targets of current treatment of CPVT, which includes the administration of β blockers, flecainide to reduce RyR2 opening, L-type Ca^2+^ channel blockers such as verapamil and use of an implantable cardioverter defibrillator (ICD) [3,4,8,68]. Flecainide, however, also blocks the Na^=^ channel by entering the pore during the open state and getting trapped there preventing further opening [69]. On the other hand, some studies indicated that flecainide does not block RyR2 but instead reduces Ca^2+^ entry by reducing the myoplasmic Na^=^ content thereby enhancing NCX extrusion of Ca^2+^ [70]. In the model, decreasing intracellular Na^+^, decreases intracellular Ca^2^ by increasing extrusion through NCX. This was shown in the previous study on alternans in a rat ventricular myocyte model by these authors [21].

## 5. Conclusions

Many experiments are conducted in mice to understand CPVT because they are mutable to transgenic expression of calsequestrin mutants. This study was performed in the guinea pig, which has a long action potential unlike the mouse, which is more similar to the human action potential. Experiments in mice show that with low pacing rates, there is an increased probability of spontaneous release events late in the action potential compared to wild type [33,71]. These experiments also show that with β adrenergic stimulation the number of late release events increases in the mutant more than in the wild type. This behavior is also seen in the computational model will mutant myocytes display late release events leading to EADs that not present in the wild type. At higher pacing rates with β adrenergic stimulation, experiments in transgenic mice with Calsequestrin mutants arrhythmia in for form of calcium alternans have been observed [72]. The model also produces this type of behavior. In humans during exercise or stimulation, CVPT has been associated with t-wave alternans [73]. Calcium alternans has been shown to underlie t-wave alternans [74]. Therefore, the mechanisms identified in the model can provide insight into clinical manifestations of CPVT.

Ca^2+^ dynamics play a major role for the heart to beat properly. The mutation in the CASQ2 expressing genes affects the Ca^2+^ dynamics by extending the opening probability of RyR2 channels with the increase in luminal dependence. Based on these simulation results, these simulation studies were able to present that alternans and EADs are the main underlying mechanisms to generate arrhythmia in autosomal dominant CPVT2.

A subcellular disruption in Ca^2+^ dynamics causes a weaker/stronger beat, longer/shorter diastolic interval, smaller/larger I_ncx_ and an increase/decrease in Ca^2+^-dependent inactivation which all affect the SR Ca^2+^ alternately. When the larger release of SR Ca^2+^ occurs to the cytoplasm, it ends up creating a stronger beat while smaller SR Ca^2+^ transients produce a weaker beat developing alternans. It was also confirmed by almost double numbers of Ca^2+^ sparks were discharged in a stronger beat than the weaker beat. A beat-to-beat change in SR Ca^2+^ load give rise to Ca^2+^ alternans which, in turn, result in cardiac alternans and APD alternans. Due to the limitations in our model, model simulations are unable to explain this mechanism by Ca^2+^ waves or mini waves.

Though EADs are also to be the precursor of arrhythmia both in tachycardia and bradycardia because it seems phase 2 EADs prefer low pacing rather than high pacing. Our simulation showed that the heart with mutant myocyte sprints with low pacing to high pacing back and forth, it can generate arrhythmia and the EADs are the mechanism behind it. These EADs generate due to the spontaneous Ca^2+^ release from the loaded SR. The late reactivation of L-type channels and spiking in electrogenic Na^+^-Ca^2+^ exchange current also aid to generate them.

## Figures and Tables

**Figure 1 genes-14-00023-f001:**
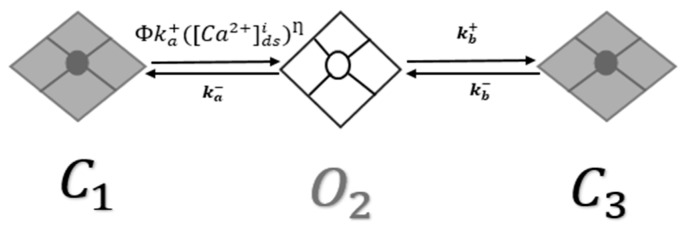
Opening probability (P_o_) of RyR2 channels from closed state (C_1_) to open state (O_2_), is controlled by luminal regulation function (Φ) in the RyR2 model.

**Figure 2 genes-14-00023-f002:**
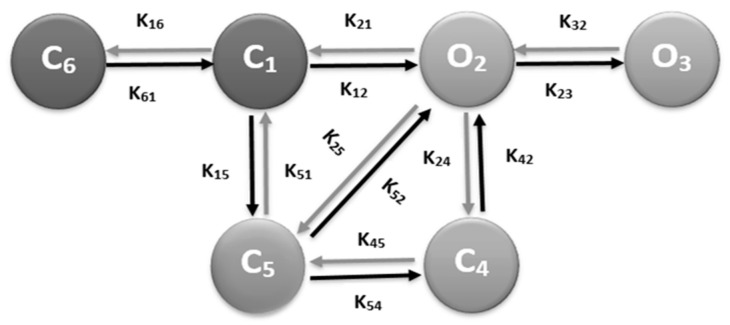
Schematic diagram of the 6-state Markov model of the L-type Ca^2+^ channel.

**Figure 3 genes-14-00023-f003:**
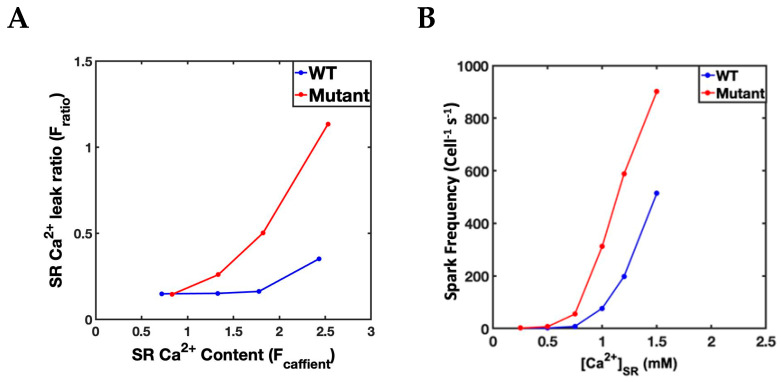
A comparison of the SR luminal dependence of mutant and wild-type myocytes Ca^2+^. (**A**) Experimentally measured values from Knollman and co-workers [32] comparing wild-type mice ventricular myocytes to transgenic mice ventricular myocytes harboring the CPVT mutation. (**B**) Experimental simulation with CPVT mutations detailed in Table 1 show that the Ca^2+^ spark frequency in simulated Guinea pig ventricular myocytes follows a similar pattern to the experiments in mice.

**Figure 4 genes-14-00023-f004:**
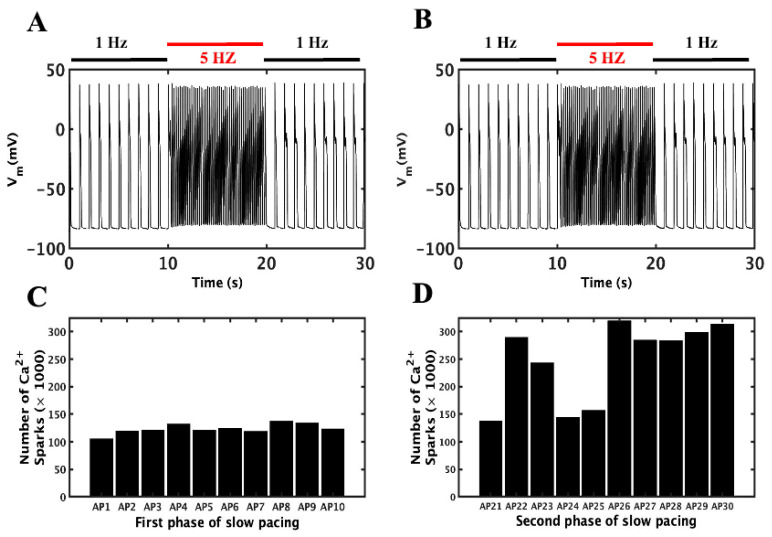
Slow–rapid–slow pacing in a CASQ2^G112+5X^ mutant myocyte generates EADs in the second phase of slow pacing during β-adrenergic stimulation. (**A**) β-adrenergic action potential simulation for 30 s in the mutant myocyte (1–10 s first phase slow pacing, 10–20 s rapid pacing, and 20–30 s second slow pacing phase). (**B**) β-adrenergic stimulation in WT myocyte with first and second slow phases with rapid pacing in the center. For the simulation in A, the number of Ca^2+^ sparks were counted and compared in the first phase slow pacing (**C**) and second phase slow pacing (**D**) after the rapid pacing. The huge difference in their number of sparks before and after the rapid pacing represents the increased SR load right after the rapid pacing. A two-tailed Student’s *t*-test unequal variance finds that the mean number of sparks are unequal in panels (**C**,**D**) with *p* < 0.001. The simulation was repeated 10 times with different seeds for the pseudo random number generator and the patterns observed were repeated each time. Shown here is a representative simulation.

**Figure 5 genes-14-00023-f005:**
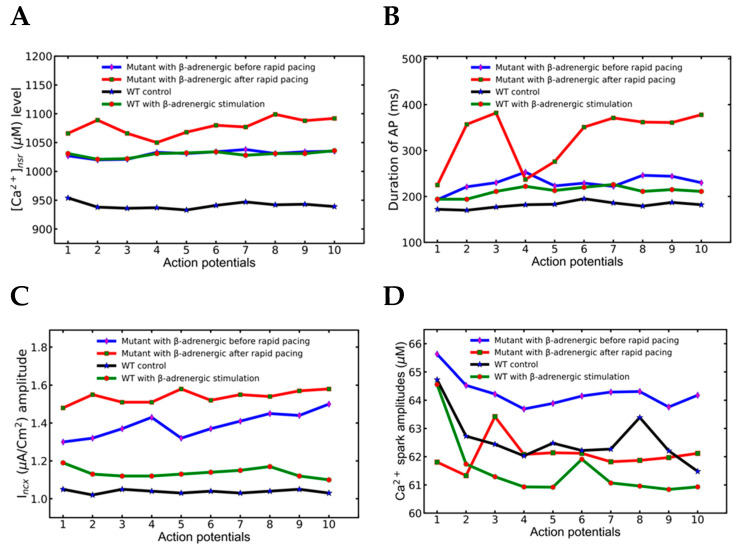
A comparison of intracellular Ca^2+^ activities of first and second slow phases of the mutant myocytes under β-adrenergic receptor with WT myocyte with or without β-adrenergic receptor-stimulated. The legend in each panel (**A**–**D**) indicates which simulation corresponds to which trace. The simulation was repeated 10 times with different seeds for the pseudo random number generator and the patterns observed were repeated each time. Shown here is a representative simulation.

**Figure 6 genes-14-00023-f006:**
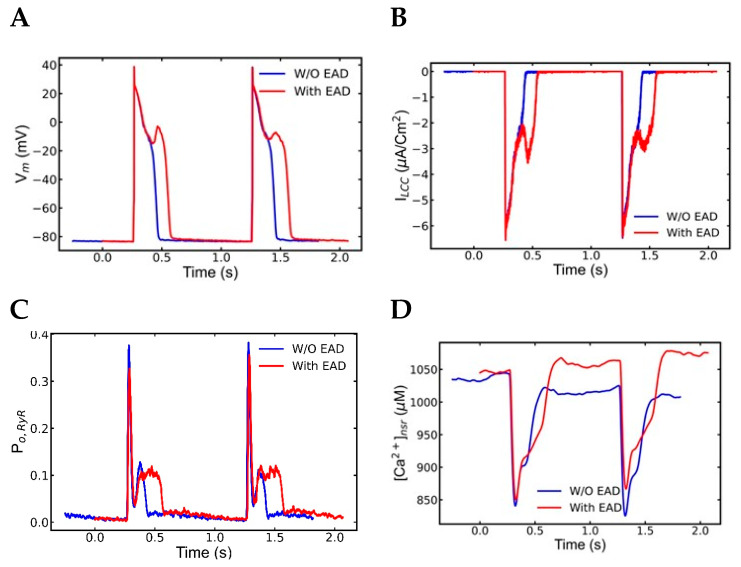
A comparison of AP, channel gating, Ca^2+^ transients, and ionic currents during β-adrenergic receptors activated myocytes show spontaneous Ca^2+^ release develops EADs in the second slow pacing after myocyte went through rapid pacing. The plots represent the combined simulations of 7.5 s to 9.5 (blue) s and 21.5 s to 23.5 s (red). (**A**) A normal action potential in the slow pacing ahead of the rapid pacing (blue), but it changed in the subsequent slow phase right after the rapid pacing where action potential developed EADs. (**B**) L-type channel current. (**C**) The RyR2 channels spontaneously release Ca^2+^. (**D**) High [Ca^2+^]_SR_ is responsible for spontaneous Ca^2+^ release. (**E**) Cytoplasmic Ca^2+^ level ([Ca^2+^]_myo_) changes after spontaneous Ca^2+^ release and late reactivation of L-type Ca^2+^ channels. (**F**) An increased activity of Na^+^-Ca^2+^ exchange current (I_ncx_) is supportive of elongate APD. The simulation was repeated 10 times with different seeds for the pseudo random number generator and the patterns observed were repeated each time. Shown here is a representative simulation.

**Figure 7 genes-14-00023-f007:**
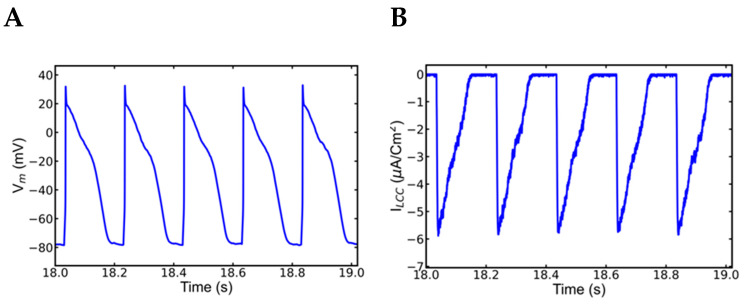
No EADs or alternans were recorded in 5 Hz pacing after the first slow phase pacing in mutant myocyte. A segment between 18 to 19 s was enlarged from Figure 3A to find more about the state of APs and other ionic currents during rapid pacing but none of them captured any abnormality. (**A**) action potentials (**B**) L-type channel current, (**C**) Na^+^-Ca^2+^ exchange current (I_ncx_), (**D**) Myoplasmic Ca^2+^ concentration ([Ca^2+^]_myo_) (**E**) NSR Ca^2+^, [Ca^2+^]nsr, and (**F**) RyR2 opening. The simulation was repeated 10 times with different seeds for the pseudo random number generator and the patterns observed were repeated each time. Shown here is a representative simulation.

**Figure 8 genes-14-00023-f008:**
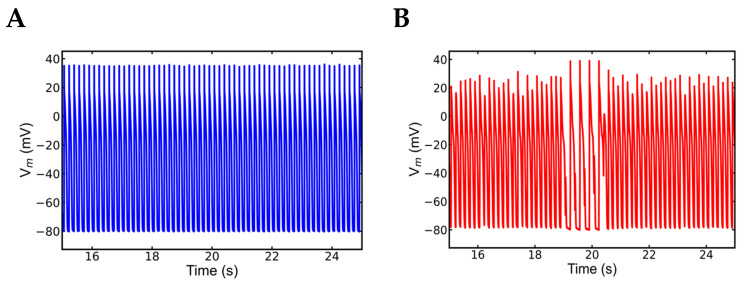
Alternans and alternate beat skipping cause arrhythmia in a myocyte having mutation in the gene expressing CASQ2 protein. (**A**) Simulation of WT myocyte with 6 Hz rapid pacing. (**B**) Arrhythmia observed in the forms of alternans and beat skipping in a mutant myocyte in rapid pacing of 6 Hz simulated with β-adrenergic stimulation. The simulation was repeated 10 times with different seeds for the pseudo random number generator and the patterns observed were repeated each time. Shown here is a representative simulation.

**Figure 9 genes-14-00023-f009:**
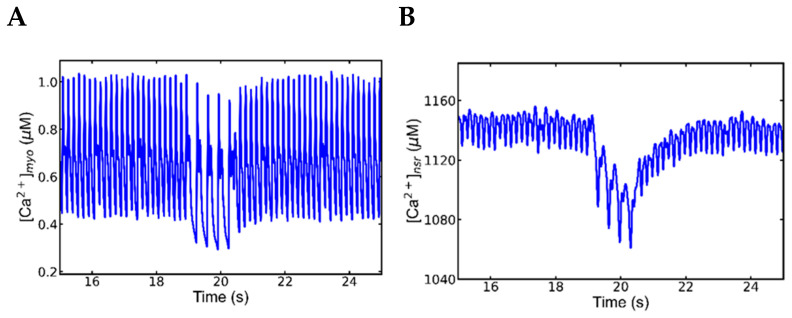
The Ca^2+^ transients, channel openings and ionic currents also reflect the patterns of alternans and the beat skipping in them. (**A**) Myoplasmic Ca^2+^ concentration ([Ca^2+^]_myo_), (**B**) NSR Ca^2+^ concentration ([Ca^2+^]_nsr_), (**C**) L-type Ca^2+^ channels, (**D**) Na^+^-Ca^2+^ exchange current (I_ncx_), (**E**) RyR2 openings, (**F**) Na^+^ current (I_Na_). The simulation was repeated 10 times with different seeds for the pseudo random number generator and the patterns observed were repeated each time. Shown here is a representative simulation.

**Figure 10 genes-14-00023-f010:**
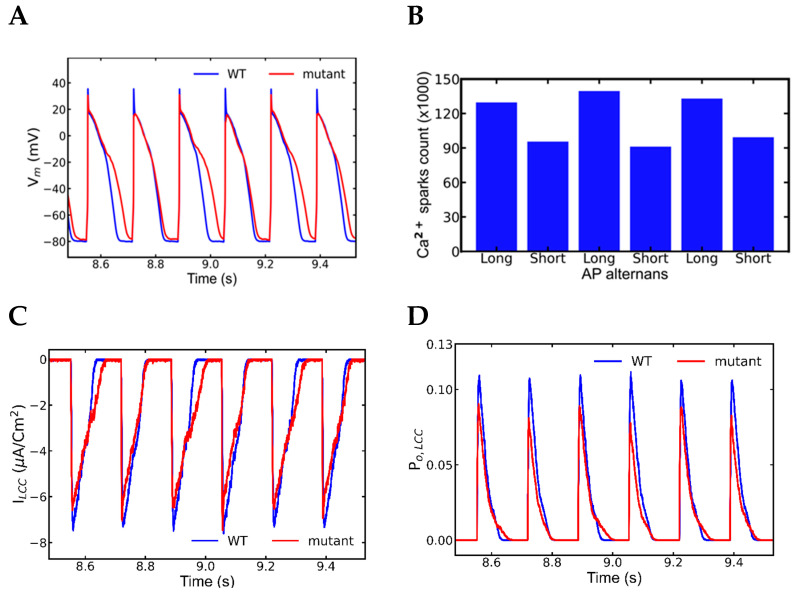
Alternate APs and Ca^2+^ dependent inactivation in alternate L-type current. APs displayed alternations in duration, amplitude, and Ca^2+^ sparks (wild-type—blue, mutant—red). (**A**) Action potential plots in between 8.5 to 9.5 s, (**B**) bar plot of Ca^2+^ sparks recorded in each beat. A two-tailed student’s *t*-test with unequal variances finds that the means of the number of sparks in the long and short beats are unequal with *p* < 0.001. (**C**) Alternate L-type current in the opposite pattern of action potential (**D**) Opening probability of L-type current displayed different pattern than L-type Ca^2+^ channel. (**E**) An open fraction of L-type Ca^2+^ channel states displayed higher calcium-dependent inactivation than voltage-dependent inactivation. The simulation was repeated 10 times with different seeds for the pseudo random number generator and the patterns observed were repeated each time. Shown here is a representative simulation.

**Figure 11 genes-14-00023-f011:**
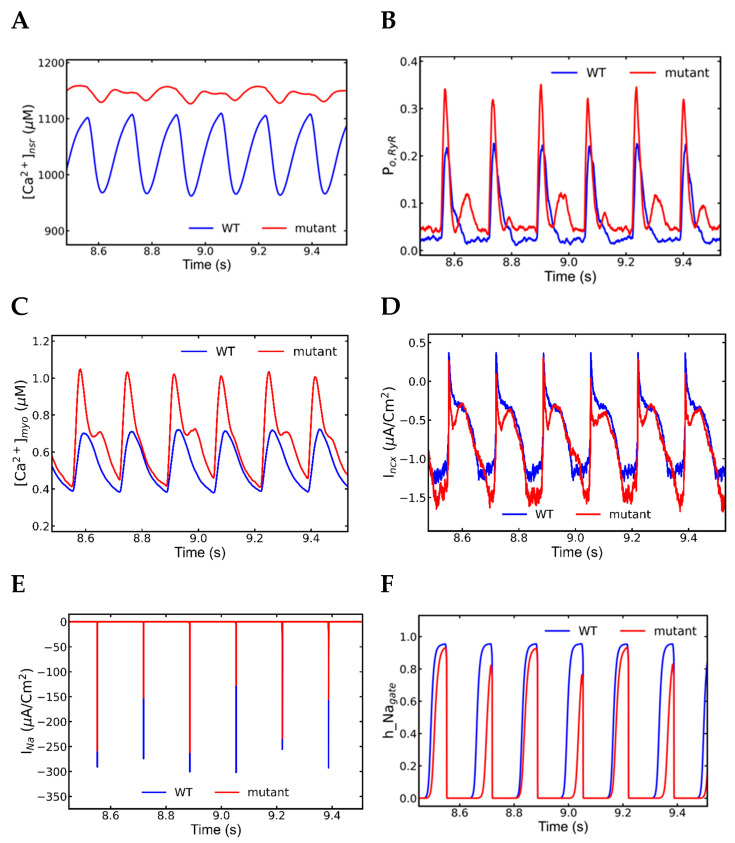
Alternate ionic currents and Ca^2+^ transients can have an alternate pattern. A detailed plots from a segment 8.5–9.5 sec displayed those patterns in different ionic elements (wild-type—blue, mutant—red). (**A**) NSR Ca^2+^ concentration (**B**) Opening probability of RyR2 channels, (**C**) Myoplasmic Ca^2+^ concentration ([Ca^2+^]_myo_), (**D**) Na^+^-Ca^2+^ exchange current (I_ncx_), (**E**) Na^+^ current (I_Na_), (**F**) Na^+^ inactivation h-gate. The simulation was repeated 10 times with different seeds for the pseudo random number generator and the patterns observed were repeated each time. Shown here is a representative simulation.

**Figure 12 genes-14-00023-f012:**
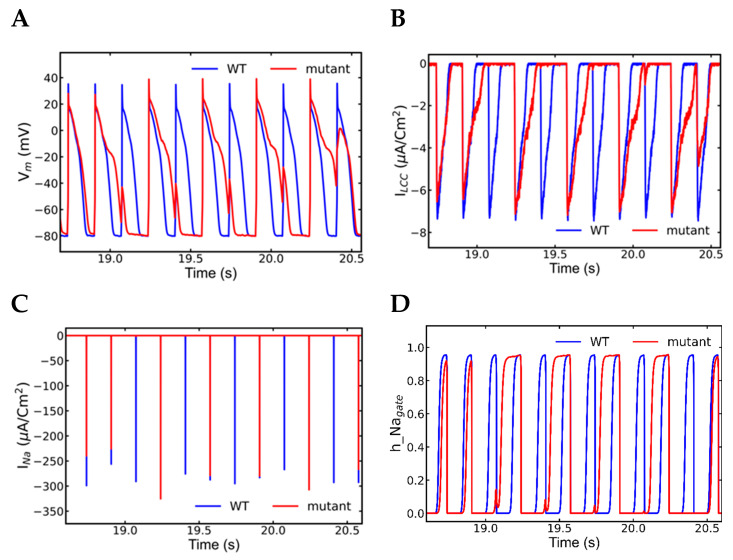
Simulation of action potential and other ionic currents in 6 Hz pacing of a mutant myocyte with β-adrenergic stimulation displayed alternately beat missing (wild-type—blue, mutant—red). The alternation in the beats happens due to the inactivation of Na^+^ channels. (**A**) Inactivation of Na^+^ channels is responsible for alternate action potential with alternately missing beats. (**B**) L–type (I_Ca_) channels are also alternately activated because they need Na^+^ current, I_Na_ (**C**) to increase membrane voltage to activate them. Na^+^ channels inactivate themselves in the peak action potential by closing their inactivation gates (**D**). Rapidly paced myocyte has a higher level of intracellular Ca^2+^ and the late resurgence of Na^+^ occurs by removing excess Ca^2+^ by I_ncx_ (**E**) and prevents Na^+^ channels to recover from the previous inactivation. When SR Ca^2+^ level diminishes (**F**), then myocyte gets back to regular beating. The simulation was repeated 10 times with different seeds for the pseudo random number generator and the patterns observed were repeated each time. Shown here is a representative simulation.

**Table 1 genes-14-00023-t001:** Simulation parameters for β-adrenergic stimulation in WT and mutant myocyte This is a table.

Parameters	Change (%)	Experimental Change (%)
Ca^2+^ regulation coefficient $\Phi_{m}$	90	Adjusted to match [32]
SERCA Pump (Ap)	50	25–50 [32,34]
JSR Volume	50	50 [32]
NSR Volume	50	50 [32]
CASQ Buffer	(−95)	−98 [32]

**Table 2 genes-14-00023-t002:** Summary of Simulation Protocols.

Simulations	Wild-Type Myocyte	Mutant Myocyte
Control Simulation	Original model parameters with a change in the pacing frequency	Morphological changes (SR Volume Increase, SR Ca^2+^ buffering diminished)
Β-adrenergic stimulation	Increase L-type activity, SERCA activity, and luminal Ca^2+^	Morphological changes plus β-adrenergic stimulation

**Table 3 genes-14-00023-t003:** Properties of Calcium Transients at 1 Hz Pacing.

**Simulations**	**Wild-Type Myocyte**	**Mutant Myocyte**	**Wild-Type Myocyte with β-Adrenergic Stimulation**	**Mutant Myocyte with β-Adrenergic Stimulation**
Ca^2+^ transient maximum (µM)	0.6889 ± 0.0076	0.7368 ± 0.0066 *	0.8417 ± 0.0271 *	0.8167 ± 0.0082 *
Diastolic [Ca^2+^] (µM)	0.1273 ± 0.0017	0.1230 ± 0.0017 *	0.1561 ± 0.0036 *	0.2194 ± 0.2709 *
**Experiment**	**Wild-Type Myocyte**	**Mutant Myocyte**	**Wild-Type Myocyte with β-Adrenergic Stimulation**	**Mutant Myocyte with β-Adrenergic Stimulation**
Ca^2+^ transient Maximum (F_ratio_)	2.23 ± 0.36	2.42 ± 0.37 *	3.22 ± 0.57	3.04 ± 0.57 *
Diastolic [Ca^2+^] (F_ratio_)	1.50 ± 017	1.70 ± 0.24	1.49 ± 0.16	1.67 ± 0.22

Values shown as averages ± standard deviation. * Significantly different from wild-type value with *p* < 0.05.

**Table 4 genes-14-00023-t004:** APD, average Ca^2+^ sparks count, and their amplitudes in myocytes (1 Hz).

Myocyte	APD (Beat) ^1^	Ca^2+^ Sparks/s ^2^	Ca^2+^ Sparks/Beat ^1^	Spark Amp. (Beat) ^1^
WT Control	181.3 ± 6.96	60,951 ± 2439	50,695 ± 1986	62.6 ± 0.85
WT β-adrenergic	211.7 ± 10.1	156,302 ± 9592	112,388 ± 5480	61.52 ± 10.7
Mutant First slow phase with β-adrenergic	229.1 ± 15.93	155,082 ± 8638	123,847 ± 8638	64.27 ± 0.52
Mutant Second slow phase with β-adrenergic	330 ± 56.95	368,841 ± 26,994	247,392 ± 68,966	62.07 ± 0.51

^1^ These values are calculated by averaging over 10 beats. ^2^ These values are calculated by averaging over 10 s.

**Table 5 genes-14-00023-t005:** Action potential amplitudes, duration, and number of sparks in alternate beats.

AP	Beat 1	Beat 2	Beat 3	Beat 4	Beat 5	Beat 6	St. Dev.
Duration (ms)	157	131	156	132	156	139	11.45
Amplitude (mV)	31.15	17.23	31.51	16.83	31.51	17.31	7.14
Spark Count	129,638	92,570	139,542	91,152	132,970	99,334	20,219

**Table 6 genes-14-00023-t006:** Alternating ionic currents and transients in consecutive beats.

Currents/Transients	Beat (n − 1)	Beat (n)	Beat (n + 1)
I_Na_	Longer	Shorter	Longer
I_Ca_	Shorter	Longer	Shorter
AP	Longer	Shorter	Longer
[Ca^2+^]_nsr_	Longer	Shorter	Longer
I_ncx_	Longer	Shorter	Longer

## Data Availability

Model codes are available as specified above in Appendix A.

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
