# Peer review of "Pacing Dynamics Determines the Arrhythmogenic Mechanism of the CPVT2-Causing CASQ2G112+5X Mutation in a Guinea Pig Ventricular Myocyte Computational Model"

_genes, 2022, doi:10.3390/genes14010023_

Round 1

Reviewer 1 Report

In this study, the authors attempted to elucidate whether the pacing dinamycs if the stimulation has any relevance in the arrythmogenic mechanism of CASQ2G112+5X mutation-related CPVQ2 in a guinea pig ventricual myocyte model. Although they conclude that according to their results there is a relation between the pacing dynamics ant the arrhtmogenics, there are some relevant questions to clarify.

The research question and relevance of the work is not completely clear. It is hard to understand which is the purpose  of the study and which can be the relevance and novelty of the study both in the field and in the clinic. 

Methodology is poorly described. The animal model is not explained. The way the achieve the ventricular myocytes for the data misses. Do they use langhendorff? Furthermore, how do they record all the currents and AP as well as calcium sparks and measurments?. Do they use patch-clamp? Do they use confocal microscope for calcium sparks? Which dye do they use for calcium measurments? Which is the protocol for currents and action potential measurments? Sometimes it seems that they base their results in in silico simulation, but other times it is not clear. Can the autors clarify this? And if it would be in silico, where does their data came from? How do the perfomr B-adrenergic stimulation?

Graphs must be improved. Data is not clearly shown. Comparations must be done between WT and mutated in the same graph. Non-concordances have been observed. For example, they mention a 1 Hz stimulation during 10 secons, which is 1 stimulus per second, but in the graph (3A) during the first 5 seconds, 10 AP are observed when we will expect 5. Figure 9E does not correspond with other figures in size, font and format. Why currents such as INA or INCX among others are expressed in uA/cm2 and not in mV? I would aldo recomend to mark in the graph changes in the pacing to make it easier to identify. 

Results do not show statistic significance, and as mention before, are hard to compare with WT and control groups. 

Discussion and conclusions answer the aim of the study, but limitations of the study as well as further impact of the study will be appreciated. Anyway, the correlation between the B-adrenergic stimulation causing CPVT2 and the pacing should be justified and explained better. 

Reviewer 2 Report

The study objective to investigate mechanism of arrhythmogenic episode in mutant CASQ2s employing simulation and stochastic model of the Guinea pig ventricular myocyte is interesting. However, the functional changes caused by the CASQ2 mutation versus WT as characterized using the rapid pacing and reverse model studies as shown by the authors are very poorly described. The authors must improve upon the following comments.

General comments:

1)     The development of the mutation (CASQ2G112+5X 15 is a 16 bp (339-354) deletion CASQ2 mutation) in the Guinea pig animals for this study is not given. This is the main model used in the study and that is not clearly described.

2)     The methodology section lacks lot of details with respect to methods of isolation of Ventricular myocytes for the study and methods/instrumentation of the calcium spark measurements.

3)     The abstract gives a feeling of study is in vivo, however, the paper to this reviewer feels like an in-silico analysis and the source of the data/ acquisition methods are not clear.

4)     The Graphs for the results in the paper are not comprehensible for this reviewer as it is not clear the comparison between control and treatment or test group. It would have been helpful to mark the changes associated with the pacing dynamics to clearly identify the point of the results shown. Also, the statistics for the results shown need to be included.

5)     Overall, the clarity of the study must be brought out for the reader to clearly understand what was done and how it was done. The exact observations/result with clear demarcations of the experimental design and state the conclusions.

6)      In the Figure 1, If the comparison of the WT and Mutant were put together in one graph above and below with the labels within the graph and indicating the places that are different between WT and mutant would be easily appreciated rather than putting them as separate ones. Additionally, here if it could be shown how without B-adrenergic stimulation these two behave can be shown it will be much appreciated to see the difference after the stimulation. The it will be easily conveyed to the reader. Here the number of readings/data acquisitions for this graph should be mentioned and the following quantification graph should reflect whether they are significant statistically.

7)     Overall, the study lack clarity on many aspects. Further, limitations of the study if any should also be discussed.

Specific comments:

Line 218- The table title states an instruction for the table placement. Correct this

Round 2

Reviewer 1 Report

Thanks to the autors for taking into account most of considerations. 

However, it is still not clear to me where the data for the simulations came from. I think it might be useful to show the row data from the in vivo and a graph that compares the in vivo data with the computational data. I still think that the methodology is hard to understand and must be compared with clinical results. In my opinion, there are too many graphs about currents, AP and calcium that could be sumarized, and the differences between control and mutated might be quantified to provided more interesting information. Thus, in my opinion results and methods should be more acqurately and clearly described. 

Moreover, I encouragly suggest to provide more explanation about the role of the B-adrenergic stimulation in the CPVT both in the introduction and in the discussion. For example, why is important the findings about the pacing dynamics in the arrithmogenics? How does it correlate with clinical?  
